# Mechanistic Chromatographic Column Characterization for the Analysis of Flavonoids Using Quantitative Structure-Retention Relationships Based on Density Functional Theory

**DOI:** 10.3390/ijms21062053

**Published:** 2020-03-17

**Authors:** Bogusław Buszewski, Petar Žuvela, Gulyaim Sagandykova, Justyna Walczak-Skierska, Paweł Pomastowski, Jonathan David, Ming Wah Wong

**Affiliations:** 1Department of Environmental Chemistry and Bioanalytics, Faculty of Chemistry, Gagarina 7, 87-100 Torun, Poland; sagandykova.gulyaim1@gmail.com; 2Interdisciplinary Centre for Modern Technologies, Nicolaus Copernicus University, Wileńska 4, 87-100 Torun, Poland; walczak-justyna@wp.pl (J.W.-S.); pawel_pomastowski@wp.pl (P.P.); 3Department of Chemistry, National University of Singapore, 3 Science Drive 3, Singapore 117543, Singapore; petar.zuvela@nus.edu.sg (P.Ž.); jonathan.david14@sps.nus.edu.sg (J.D.)

**Keywords:** RP-HPLC, mixed-mode HPLC, QSRR, flavonoids, antioxidant activity, mechanistic study

## Abstract

This work aimed to unravel the retention mechanisms of 30 structurally different flavonoids separated on three chromatographic columns: conventional Kinetex C18 (K-C18), Kinetex F5 (K-F5), and IAM.PC.DD2. Interactions between analytes and chromatographic phases governing the retention were analyzed and mechanistically interpreted via quantum chemical descriptors as compared to the typical ‘black box’ approach. Statistically significant consensus genetic algorithm-partial least squares (GA-PLS) quantitative structure retention relationship (QSRR) models were built and comprehensively validated. Results showed that for the K-C18 column, hydrophobicity and solvent effects were dominating, whereas electrostatic interactions were less pronounced. Similarly, for the K-F5 column, hydrophobicity, dispersion effects, and electrostatic interactions were found to be governing the retention of flavonoids. Conversely, besides hydrophobic forces and dispersion effects, electrostatic interactions were found to be dominating the IAM.PC.DD2 retention mechanism. As such, the developed approach has a great potential for gaining insights into biological activity upon analysis of interactions between analytes and stationary phases imitating molecular targets, giving rise to an exceptional alternative to existing methods lacking exhaustive interpretations.

## 1. Introduction

Flavonoids as secondary plant metabolites perform various functions in growth, development, reproduction, and abiotic responses [1]. The structural diversity, biological and ecological significance, and health-promoting and anti-cancer properties of flavonoids have been attractive for scientists from different disciplines [2]. Special attention to flavonoids is paid by analytical chemists because of their bio- and nutraceutical-activity, as well as benefits for dietary supplementation. Therefore, there is increased interest in the advancement of analytical techniques for plant extract and food quality analysis. However, the structural diversity of flavonoids can also lead to challenges in separation and, thereby, their analysis. It was estimated that, in 2003, more than 9000 flavonoid derivatives were reported. Based on the possible substitution patterns of ten carbon atoms comprising the flavonoid skeleton, the number of theoretically-viable structures is monumental [2].

Reverse-phased high-performance liquid chromatography (RP-HPLC) has proven to be a suitable method for the analysis of flavonoids due to their aglycone structure, nature, and degree of glycosylation and acylation [3]. Therefore, understanding the retention mechanism and interactions of flavonoids with the chromatographic phases governing the retention is of great concern to enhance selectivity and reduce the costs of analysis by shortening the number of required experiments for method development.

Quantitative structure retention relationships (QSRRs), introduced by the pioneering research of Professor Roman Kaliszan, relate solute retention and their molecular structure [4]. Since the inception of QSRRs in the early 1970s, numerous applications have been reported, such as (i) prediction of retention time [5], (ii) estimation of the lipophilic character of analytes [6,7], (iii) determination of biological activities of analytes [8,9], (iv) metabolite identification in non-targeted metabolomics [10], and (v) columns characterization and selection [11,12,13].

In the case of flavonoids, QSRRs have been mostly employed for prediction of retention time and estimating their lipophilic character, as outlined in a recent review [14]. Since the key drawback of QSRR modeling of flavonoids is lack of exhaustive interpretations (i.e., ‘black box’ approach), notable improvements can still be made. With the rapid development of computational methods and thus the increase in the number of available molecular descriptors, risk of over-fitting and chance correlation increases and can be avoided using appropriate regression methodology in combination with variable selection, leading to improved accuracy and robustness of QSRR models [15,16]. However, many molecular descriptors are difficult to physically interpret or have no physical meaning whatsoever. Mechanistic interpretations are not only important for the prevailing QSRR applications but may also be employed for evaluation of biological activity if the stationary phase imitates a molecular target.

Quantum-mechanical (QM) descriptors calculated using density functional theory (DFT) methods allow for insights into retention mechanisms at the molecular level [17]. Since DFT calculations can be time-consuming, whereas the correlations of QM-derived descriptors with retention time are not always satisfactory [18], their application in QSRR retention mechanism modeling is challenging.

For instance, Akbar et al. [19] used QM descriptors together with 18 blocks of 2D and 3D descriptors to build robust stepwise multiple linear regression (MLR) QSRR models for the prediction of retention times of naturally-occurring flavonoids. The authors based their conclusions about the RP-HPLC retention mechanism on vague interpretations, while QM descriptors failed to be included in the QSRR models.

Conversely, Zapadka et al. [6] comprehensively assessed lipophilicity (expressed as log*k*_w_ [20]) of a series of flavonoids using QSRRs on two chromatographic columns—Synergy POLAR and Synergy-FUSION in the RP mode together with MLR, molecular properties (e.g., in silico log*P*), and 3D-descriptors. However, the application of selected descriptors for two QSRR models allowed for the identification of the structural features, governing the retention without the insights into the underlying molecular mechanisms. The largest contributing parameter was predictably found to be in silico log*P*, introducing considerable redundancy to the model.

Tache et al. [21] reported yet another approach for insights into chromatographic interactions governing the separation of flavonoids on different stationary phases, namely, chemically-bonded silica phases with highly end-capped octadecyl, polar embedded linker octadecyl, phenyl, and pentafluorophenyl ligands. Chromatographic behavior of these stationary phases was evaluated by means of graphical profiles and correlation matrices, lipophilicity charts, and principal component analysis (PCA) loading plots. Despite the given insights into interactions of analytes with the stationary phases, such an approach was more suitable for the estimation of lipophilic character since the interactions themselves were not thoroughly discussed.

In this work, we presented a novel approach for predicting the retention of 30 structurally-different flavonoids on three chemically-bonded stationary phases (classical octadecyl chain, pentafluorophenyl, and diacylated phosphatidylcholine) using genetic algorithms–partial least squares QSRR modeling [10,16,22] based on DFT-derived QM molecular descriptors. Mechanistic interpretations were given for the first time based on a comprehensive analysis of underlying relationships between the DFT-derived QM descriptors and retention inferring valuable physicochemical meaning.

## 2. Results and Discussion

### 2.1. HPLC-MS/MS Analyses

Triple quadrupole mass spectrometry allowed for the identification of retained flavonoids via specific multiple reaction monitoring (MRM) transitions without the need for comprehensive separation. Some representative MRM transitions and retention times for all the columns are depicted in Figure 1, whereas the remaining chromatograms can be found in the Appendix A (Appendix A and retention times in Appendix A). Co-eluting compounds with similar MRM transitions were identified by additional analysis of standards. Results of the chromatographic analyses showed that different peak shapes could be observed across three stationary phases. For instance, peaks of fisetin, scutellarein, and myricetin, when separated on the K-C18 column (Figure 1A), exhibited significant tailing. Peaks of the same analytes separated on the K-F5 column (Figure 1B) had a shape closer to the Gaussian distribution. The noisiness of the scutellarein peak (Figure 1C) likely originated because of its low retention on the immobilized artificial membrane (IAM) column, and its retained concentration was on the border of limit of detection (LOD). Additional peaks for scutellarein (Figure 1B) and myricetin (Figure 1A,B) were artifacts and did not correspond to the presence of another conformation.

However, fisetin and scutellarein still exhibited minor peak tailing. On the other hand, when separated on the IAM.PC.DD2 column, these flavonoids exhibited a broad shape (Figure 1C). Despite the discussed differences across the three columns, it is worth noting that on an experimental scale, the elution order for most of the analytes was similar: (1) K-F5, (2) K-C18, and (3) IAM.PC.DD2 except for hesperetin (with quite similar retention times for K-C18 and IAM.PC.DD2). Other exceptions included wogonin, pinocembrin, pectolinarigenin, 5-hydroxyflavone, 3,5-dihydroxyflavone, tectochrysin, and, for them, the elution order across the stationary phases was: (1) K-F5, (2) IAM.PC.DD2, and (3) K-C18.

Upon transforming the retention time for all the three chromatographic columns to the natural logarithm scale, their respective values followed a statistical distribution with a considerably higher degree of normality.

Similarities/dissimilarities between K-C18, K-F5, and IAM.PC.DD2 columns were initially evaluated via simple (Pearson) correlation analysis with a two-tailed significance test. As could be observed from Table 1, all the retention times (*t*_R_) were significantly correlated (*p* < 10^−5^) across the three evaluated columns. The conventional K-C18 and K-F5 columns exhibited the strongest correlation, and thereby the highest degree of similarity. The IAM.PC.DD2 column exhibited the lowest correlation with both K-C18 and K-F5 columns (*R* < 0.8, *p* < 10^−5^), and it was thereby the least similar. The dissimilarities could be attributed to the dual nature of the phosphatidylcholine (PC) ligand of the IAM.PC.DD2, a lipid with a long hydrophilic chain (tail) and a hydrophobic head comprising of two charged centers.

### 2.2. Consensus Genetic Algorithm-Partial Least Squares (GA-PLS) QSRR Model for the K-C18 Column

Results of GA hyper-parameter optimization showed that the minimum root mean square error of cross-validation RMSECV values were obtained for a population size of 20, cross-over fraction of 0.8, and the mutation rate of 0.2. The number of latent variables (LVs) was optimized within each unit of a GA population using leave-one-out cross-validation (LOO-CV) and was reported separately for each chromatographic column.

#### 2.2.1. Selected Molecular Descriptors and Predictive Ability for the Consensus K-C18 GA-PLS QSRR Model

Final consensus QSRR model for the K-C18 column after 1000 GA-PLS runs comprised of four variables, namely, in the order of decreasing occurrence (Figure 2A): the number of hydroxyl groups (*n*(OH)), the total dipole moment (M_tot._), solvation energy (SE), and the ionization potential (IP). The minimum bond dissociation enthalpy of the first oxidation step of the hydrogen atom transfer (HAT) mechanism (BDE_min_) was oddly not included in the final model and, as such, did not affect retention of flavonoids on the K-C18 stationary phase. Nevertheless, it was accounted for through the number of OH groups to which it was significantly inversely correlated (*R* < −0.7, *p* > 10^−5^). Other parameters describing electrostatic interactions between the analytes and both of the chromatographic phases were less represented.

Besides the GA parameters, the number of LVs was carefully optimized (Appendix A). The optimal number of LVs for the K-C18 consensus GA-PLS QSRR model was found to be three with an RMSECV of 2.39 min (0.190 on the ln scale). Three LVs explained 98.1% of the variance in X-space (molecular descriptors), and 92.6% of the variance in Y-space (retention time). The developed QSRR model exhibited strong predictive ability, as evident from Figure 2B, on both the training and the validation sets. The resulting average RMSE was 1.95 min (0.164 on the ln scale), with a slightly higher error of the training set, which could be attributed to the lower number of compounds analyzed in this study (30 flavonoids). Finally, the consensus model was found to be strongly statistically significant (Table 2) with an *F* value of 70.28 and a *p*-value < 10^−3^.

#### 2.2.2. Mechanistic Interpretations of the K-C18 Consensus GA-PLS QSRR Model

According to the PLS coefficients (Figure 2C), the most influential variables were SE and *n*(OH), while the influence of the molecular descriptors encoding the electrostatic interactions between flavonoids and the two chromatographic phases was considerably less pronounced.

It was not surprising that SE was found to be the most important variable as it accounted for both of the key driving forces that dominate the RP-HPLC retention mechanism involving the octadecyl-bonded stationary phases: hydrophobicity (typically expressed as log P) and solvent effects. Solvent effects expressed through the dispersive (van der Waals) interactions between the analytes and the chromatographic phases are typically approximated using the solvent-accessible surface area (SASA) [23]. Levy et al. [24] argued that due to the poor transferability of the surface area models, a single SASA parameter was insufficient to reproduce the solute-solvent van der Waals energies. Thereby, SE is a better parameter to account for not only hydrophobic interactions but also the effects of dispersion within the K-C18 retention mechanism because the solvation model based on density SMD solvation model comprises of electrostatic and cavity-dispersion-solvent-structure terms [25]. With the increase of SE (i.e., the difficulty of solvating the analytes), retention time also notably increased due to the positive sign and high magnitude of its corresponding PLS coefficient (Figure 2C).

The number of hydroxyl groups and total dipole moment were negatively correlated to retention time. With the increase of *n*(OH) and M_tot._ values, retention time decreased due to the increased polarity of the flavonoids, whereby they were less likely to interact with the non-polar stationary phase. Furthermore, ionization potential exhibited weaker negative correlations with retention time.

#### 2.2.3. Chemical Domain of Applicability of the K-C18 Consensus GA-PLS QSRR Model

Figure 2 shows that all of the analytes fell well within the warning limits of applicability domain (AD) (i.e., three multiples of the standard deviation of standardized residuals and critical leverage – *h** of 0.714). This readily confirmed the robustness and stability of the K-C18 QSRR model.

### 2.3. Selected Molecular Descriptors and Predictive Ability for the Consensus K-F5 GA-PLS QSRR Model

After 1000 runs of GA-PLS, a six-variable consensus QSRR model was built. The six variables comprising the model with the highest occurrence (Figure 3A) were the solvation energy (SE), number of hydroxyl groups (*n*(OH)), electron transfer enthalpy (ETE), total dipole moment (M_tot._), ionization potential (IP), and electron affinity (EA). As could be observed from Figure 4A, SE was occurring in all the models, whereas *n*(OH) and M_tot._ occurred in nearly all the models, followed by IP, ETE, and EA.

The final consensus QSRR GA-PLS model for the K-F5 was found to be statistically significant (*F* value of 49.84, and *p*-value of 3.63 × 10^−6^, Table 3) and consisted of an optimal number of four latent variables (LVs), which yielded an RMSECV of 1.49 min (0.216 on the ln scale, Appendix A). Four LVs explained 94.0 and 97.2% of the variance in X-space (molecular descriptors) and Y-space (retention time), respectively. As could be observed from Figure 3B on the training and the validation sets, the model was strongly predictive, with an average RMSE of 1.24 min (0.257 on the ln scale).

#### 2.3.1. Mechanistic Interpretations of the K-F5 Consensus GA-PLS QSRR Model

Molecular descriptors with the strongest influence on retention time for the K-F5 QSRR model were found to be the descriptors: SE, *n*(OH), ETE, and IP (Figure 3C). This confirmed the similarity between K-F5 and K-C18 columns, as evident from the initial retention time correlation analysis (*R* = 0.929, *p* = 1.42 × 10^−13^). Hydrophobicity and dispersion effects still remained the dominant interactions affecting retention, as SE exhibited the strongest positive correlation with retention time (Figure 3C).

Again, the number of hydroxyl groups and total dipole moment were negatively correlated to retention time. With the increase in *n*(OH) and M_tot._, the polarity of the flavonoids increased, which led to a decrease in the retention time. Since the employed mobile phase was polar, this implied weaker interactions with the K-F5 stationary phase, which, despite comprising of polar ligands (pentafluorophenyl), is known to exhibit both normal- and reversed-phase behavior depending on the mobile phase composition [26].

Since the pentafluorophenyl ligand of the K-F5 column is a π-electron system, it is expected to interact with aromatic solutes via π-π interactions.

Despite the fact that non-covalent interactions are usually weak, their contributions to retention cannot be neglected as their collective strength can define retention behavior. Fluorine substituents comprising the benzene ring of the K-F5 column ligands may also have an effect on the strength of π-π interactions, but experimental evidence demonstrated that acetonitrile (ACN) might suppress them [27]. Such a phenomenon may occur since ACN is an electron-rich organic modifier and tends to participate in π-π interactions with a higher affinity towards flavonoids. Preferential sorption of ACN onto the stationary phase may also occur in case when it is more electron-deficient than the solutes, thus complicating solute-stationary phase interactions [28]. Experimental evidence demonstrates this phenomenon, whereby using methanol as a mobile phase may enhance the selectivity of phenyl-based stationary phases towards aromatic solutes by enhancing the strength of π-π interactions [29]. Moreover, Emenike et al. [30] reported that the solvent notably affected CH-π interactions, and the substituent effect could be washed out for particular solvents, thus emphasizing the importance of considering solvent effects on weak non-covalent interactions. The interplay between solvent and substitution effects have also been observed for edge-to-face aromatic interactions [31].

Furthermore, the flavonoids also exhibited electron transfer behavior with the mobile and stationary phases, both donation and acceptance. This was encoded through the ETE, IP, and EA descriptors. ETE and EA exhibited strong and weak positive correlations with retention time. On the other hand, IP, the susceptibility of the flavonoids to lose electrons, exhibited a strong negative correlation. Thereby, the analytes exhibited the stronger and more immediate influence of electrostatic interactions (Figure 3C).

#### 2.3.2. Chemical Domain of Applicability of the K-F5 Consensus GA-PLS QSRR Model

The Williams plot displayed in Figure 3D represents the chemical domain of applicability (AD) for the K-F5 consensus GA-PLS QSRR model. All the analytes were found to be well within the AD limits of three multiples of the standard deviation of standardized residuals and the critical leverage value of 1.000, except tectochrysin with a high standardized residual of −3.735. This flavonoid had some specific features that contributed to its high retention time value, as it could be observed from Figure 3D.

Instead of an O-H group at the 5′ position, it had an O-CH_3_ group, while the sole O-H group (on the 3′ position) formed an intramolecular hydrogen bond with techtochrysin’s C=O group.

### 2.4. Selected Molecular Descriptors and Predictive Ability for the Consensus IAM.PC.DD2 GA-PLS QSRR Model

For the IAM.PC.DD2 column, 1000 runs of GA-PLS yielded a consensus QSRR model comprising of seven molecular descriptors. In the order of decreasing occurrence: solvation energy (SE), total dipole moment (M_tot._), number of hydroxyl groups (*n*(OH)), minimum bond dissociation enthalpy (BDE), ionization potential (IP), energy gap between the highest occupied molecular orbital and lowest unoccupied molecular orbital (HOMO-LUMO gap,ΔE_HOMO-LUMO_), and the global hardness (*η*) (Figure 4A).

The resulting consensus model was built out of four latent variables that yielded an RMSECV of 1.96 min (0.129 on the ln scale, Appendix A) and explained 91% of the variance in X- (molecular descriptors) and 93% of the variance in Y-space (retention time). Although not as strong as in the case of K-C18 and K-F5 columns, the IAM.PC.DD2 model exhibited a reasonable predictive ability with an average RMSE (over the training and validation sets) value of 1.60 min (0.099 on the ln scale). Regardless, the model was found to be strongly statistically significant (Table 4) with an *F* value of 39.39 and a *p*-value of 1.05 × 10^−4^.

#### 2.4.1. Mechanistic Interpretations of the IAM.PC.DD2 Consensus GA-PLS QSRR Model

Retention mechanisms of the K-C18 and K-F5 stationary phases were dominated by hydrophobic interactions and pronounced solvent effects. Interestingly, despite a considerable positive contribution of SE towards retention (interactions with the hydrophilic “tail” of phosphatidylcholine – PC), in the case of the IAM.PC.DD2 stationary phase, electrostatic interactions with the hydrophobic “head” of PC were also found to be dominating (Figure 4C). These findings highlighted the advantages of the study over existing works on the retention of flavonoids using the IAM.PC.DD2 column. As compared to the conventional approaches, the developed QSRR model did not exploit the log*k*_w_/log*P* and log*k*_w_/log*D* ratios for neutral and ionizable compounds, respectively. Typically, the use of such partition coefficient ratios yields good statistics, but the interpretation of retention mechanisms can be insufficient. For instance, Tsopelas et al. [32] had not considered the influence of electrostatic interactions on the retention of flavonoids on the IAM.PC.DD2 and IAM.PC.MMG columns (typically studied by introducing positively (F^+^) and negatively (F^−^) charged molecular fractions of analytes). Instead, the authors remarked that hydrogen bonding between the analytes and the chromatographic phases affected the retention of the IAM.PC.DD2 column based on only moderate correlations between log*k*_w_ values of the IAM.PC.DD2 and IAM.PC.MMG columns. Despite the authors’ focus on relating the retention behavior of flavonoids to cell permeability, retention mechanisms were poorly characterized [32].

Santoro et al. [9] analyzed the interactions of the analytes with the three IAM columns (IAM.PC.DD, IAM.PC.DD2, and cholesteryl ester) using the conventional partition coefficient ratio approach. Slopes of the corresponding retention curves allowed for the separation of analytes into two groups: hydroxylated and non-hydroxylated. The reported differences in the retention behaviors between these two groups on IAM.PC.DD and IAM.PC.DD2 columns might be attributed to the contribution of hydrogen bonding. However, even the authors themselves remarked that the use of such an approach could give rise to a wrong impression of similarity between retention mechanisms of IAM.PC.DD and IAM.PC.DD2 vs. cholesteryl ester columns based solely on statistical parameters (*Q*^2^, slopes of QSRR equations). Correlations between log*P*_oct._ and log*k*_w_ showed that for the cholesteryl ester column, predominant retention mechanism was partitioning, while, for the IAM column, there was an additional mechanism present [9].

Besides partitioning, our approach also considered polarity of the flavonoids and electrostatic interactions. It was shown that the number of hydroxyl groups and the total dipole moment were still contributing descriptors, exhibiting an intermediate negative correlation to retention time. This was not surprising since, with the decrease of the polarity of the flavonoids, they were more strongly attracted to the non-polar hydrophilic segment of PC, and thereby retained more strongly on the IAM.PC.DD2 stationary phase.

On the other hand, the minimum O-H bond strength expressed through BDE (significantly negatively correlated with *n*(OH), *R* < −0.7, *p* > 10^−5^) was found to be positively correlated with retention time. The stronger the (weakest) O-H bond was (higher the BDE), the more electron-deficient was its respective O-H hydrogen. This led to a formation of stronger hydrogen bonds between O-H hydrogen and the active charged centers of the hydrophobic “head” of PC.

The other key contributing descriptors, encoding the electrostatics of the IAM.PC.DD2 column, were the global hardness (*η*) and HOMO-LUMO energy gap, which were both strongly negatively correlated to retention time (Figure 4C), whereas the most negative excess (NBO, natural bond orbital) charge exhibited a weaker positive correlation to retention time. Accordingly, the decrease in the HOMO-LUMO gap, which resulted in an ease of charge transfer, as well as the decrease of the *η* molecular descriptor, which accounted for the resistance of flavonoids towards charge transfer, led to a considerable increase in retention time. On the other hand, the more negative was the atom with the most negative charge, the stronger was that flavonoid retained on the IAM.PC.DD2 stationary phase. Such a mechanistic description pointed to a fact that there are competing interactions for the separation of flavonoids on the IAM.PC.DD2 stationary phase, which is in excellent agreement with its dual nature.

#### 2.4.2. Chemical Domain of Applicability of the IAM.PC.DD2 Consensus GA-PLS QSRR Model

Finally, the chemical domain of applicability (AD) of the IAM.PC.DD2 consensus QSRR GA-PLS model was also defined. All the analytes were found to lay within the warning limits of the AD (three multiples of the standard deviation of standardized residuals, and the critical leverage of 1.263 (Figure 4D). Although well within the AD, one flavonoid, catechin, was found to be considerably structurally different from the others, exhibiting nearly the lowest SE value of −27.785 kcal/mol. Only epicatechin and epigallocatechin had similar values: −27.271 kcal/mol and −27.626 kcal/mol, respectively. Since both epicatechin and epigallocatechin were included in the training set, the low standardized residual for catechin was not surprising, despite its somewhat large leverage. All three belonged to the family of catechins, exhibiting notable deviations from planarity and stereoisomerism, whereas their C-ring did not contain both C=C and C=O double bond. The presence of a C=C bond within the C-ring of the flavonoids (the readers are referred to Appendix A for the 3D molecular structures) made its structure nearly planar together with the benzene ring, stabilizing it in the process. For the purposes of this study, we considered the more stable equatorial catechin isomers.

## 3. Materials and Methods

### 3.1. Reagents and Chemicals

Standards of flavonoids (genistein, scutellarein, epicatechin, kaempferol, eriodictyol, apigenin, liquiritigenin, fisetin, taxifolin, hesperetin, 3′,4′-dihydroxyflavonol, diosmetin, morin, epigallocatechin, dihydroxymyricetin, myricetin, wogonin, 7,8-dihydroxyflavone, chrysin, pinocembrin, catechin, baicalein, 3,5-dihydroxyflavone, galangin, genkwanin, 5-hydroxyflavone, tectochrysin) were purchased from Sigma Aldrich (St. Louis, MO, USA) with purities > 90%. Pectolinarigenin (min. 75% purity), 3,5,7,8,3′,4′-hexahydroxyflavone (min. 98% purity), 5,3′,4′-trihydroxyflavone (min. 98% purity) were purchased from Carbosynth (Oxford, United Kingdom). Stock solutions were prepared by dissolving them in ethanol (90%), except for scutellarein, genkwanin, 5,3′,4′ - hydroxyflavone that were dissolved in 1:1 mixture of acetonitrile-methanol (*w*/*w*) with subsequent sonication because of their very low solubility (determined in-house due to lack of literature data). The final concentration of standard solutions was 1 mg/mL. Solvents, such as methanol, acetonitrile, were of LC-MS grade purity. Fresh mixtures of analytes consisted of 30 flavonoids and were prepared by transferring 50 µL of a standard solution of each analyte into 2-mL vials, followed by thorough mixing and were applied for each column.

Coverage of the different flavonoid families analyzed in this work focused mostly on flavones and flavonols, and flavonol species. Out of a total of 30 flavonoids, 40.0% belonged to flavones, 26.3% to flavonols, 13.33% to flavanones, 10.0% to flavans, 6.7% to flavanonols, and 3.3% to isoflavones.

### 3.2. HPLC-MS/MS Conditions

Retention data of selected flavonoids was obtained in 3 replications and using 3 different chromatographic columns: Kinetex C18 (K-C18), Kinetex F5 (K-F5), immobilized artificial membrane column (IAM.PC.DD2) with 0.1% trifluoroacetic acid (TFA) in water as mobile phase A and acetonitrile as mobile phase B. Gradient program was as follows: 0.01 min − 25%, 30 min − 80%, 35 min − 80%, 40 min − 25% of mobile phase B at 30 °C for all columns. Measurements were carried out using LC-MS 8050 triple quadrupole mass spectrometer (Shimadzu, Kyoto, Japan) equipped with a binary solvent delivery system (LC-30AD), a controller (CBM 20A), an autosampler (SIL-30A), column thermostat (CTO-20AC). Chromatograms were processed using LabSolutions 5.8 software (Shimadzu, Kyoto, Japan).

Prior to the separation of a prepared mixture, MRM transitions specific for each analyte were optimized without HPLC separation. Injection volume accounted for 1 µL with a flow rate of 0.5 mL/min in positive ionization mode and collision energy 30 keV.

The settings of ESI included: nebulizing gas flow 3 L/min, heating gas flow 10 L/min, the temperature of the drying gas 400 °C, desolvation line (DL) temperature 250 °C, and interface temperature 300 °C.

### 3.3. Chromatographic Columns

Chromatographic columns studied in this work included Kinetex C18 (K-C18, Phenomenex, Torrance, CA, USA), Kinetex F5 (K-F5, Phenomenex, Torrance, CA, USA), and IAM.PC.DD2 (Regis Technologies, Morton Grove, IL, USA). Technical parameters of the selected columns are summarized in Table 5. Surface coverage (*α*) was determined according to the Berendsen-de-Galan equation [33]. According to the structures of the ligands chemically-bonded to the stationary phases (Figure 5) and the information from the manufacturer, the K-C18 column is expected to demonstrate hydrophobic interactions due to octadecyl carbon chain, while K-F5 may exhibit diverse interactions. Alongside the classical reversed-phase retention mechanism dominated by hydrophobic interactions, π-π electrons of the benzene ring could also affect retention. IAM.PC.DD2 columns are expected to exhibit a dual nature since the immobilized PC ligands contain both hydrophobic regions and charged centers, resulting in both hydrophobic and electrostatic interactions.

### 3.4. Mechanistic QSRR Model Development

Correlation analysis for retention times across all the evaluated chromatographic columns was carried out using the Pearson correlation coefficient (*R*). The statistical significance of the *R* values was tested using the two-tailed test. Retention times of all the chromatographic columns were transformed to the natural logarithm scale to change its statistical distribution to normal. Subsequently, molecular structures of 30 flavonoids were prepared according to the protocol of Žuvela et al. [34,35]. Briefly, the structures were first drawn in ChemDraw Prime 18.2 (PerkinElmer Inc., Waltham, MA, USA) and subjected to extensive conformational analysis using the molecular mechanics’ method with the Merck molecular force field MMFF [36,37] force field. The conformational analysis was paramount due to the possibility of intramolecular hydrogen bond formation within the B-ring of some of the analyzed flavonoids [38].

It comprised of three segments: (1) torsion rotation, (2) two correlated rotations to keep the rings closed, and (3) a six-member flip. After each step, the resulting structures were subjected to energy minimization. All the produced structural conformations were optimized using the semi-empirical AM1 [39] method. Twenty lowest energy structures were further refined at the HF/3-21G [40] level of theory. Five final HF/3-21G conformers were optimized by employing density functional theory (DFT) [41] with the ωb97xD [42] functional and the 6-311++G(d,p) [40] basis set. The conformational analysis was performed in vacuo.

Calculation of mechanism-specific molecular descriptors (see section Molecular descriptors for mechanistic QSRR modeling) required building and optimizing molecular structures of not only neutral but also anion, mono-radical, radical anion, and diradical species. The conformational analysis was performed on neutral, anion, and mono-radical species of all the analytes. Due to high computational costs, the radical anion and diradical species of the analytes were directly optimized at the highest level of DFT theory.

Upon completing the in vacuo conformational analysis and optimization, all the calculations were further carried out in implicit water solvent using the SMD solvation model [25] due to pronounced solvent effects of the HPLC retention mechanism. Mechanistic QSRR models were built out of 13 molecular descriptors, which represented parameters of two well-established mechanisms of antioxidant activity (hydrogen atom transfer - HAT, and sequential proton-loss electron-transfer - SPLET) [43], electrostatics (dipole-dipole, dipole-induced dipole, polar interactions, hydrogen bonding) [11,23,44,45], as well as hydrophobic and solvent effects (dispersive – London-type interactions) [23].

Prior to modeling, the Kennard and Stone algorithm [46] was employed for stratified dataset sampling into 23 training (70%) and 10 external validation (30%) analytes. Genetic algorithm-partial least squares (GA-PLS), recently shown to be effective in QSRR [10,16,22], was used for simultaneous modeling and selection of the most informative molecular descriptors. However, instead of a mixed-integer formulation (with a fixed number of selected variables), the binary implementation of GA [47,48] was employed instead. Parameters of GA-PLS (i.e., cross-over fraction, mutation rate, and the number of latent variables) were optimized using leave-one-out-cross validation (LOO-CV). Comprehensive validation of the models was performed using LOO-CV, external validation, while the models’ chemical domain of applicability was also defined. Statistical significance of the resulting models was tested using a CV-analysis of variance (CV-ANOVA) [49].

Conformational analysis and all the pertinent QM and chemometric calculations were performed using Spartan 14 (Wavefunction, Inc., Irvine, CA, USA), Gaussian 16 (Wallingford, CT, USA, Ref. S1), and MATLAB 2019a (MathWorks, Natick, Massachusetts, USA), respectively.

### 3.5. Theoretical Methods

#### 3.5.1. Partial Least Squares (PLS)

QSRR models developed in this work were built using (PLS). PLS is an exploratory analysis and regression method introduced by Wold [50,51]. The method is based on compressing the original variables (predictors, **X**, and dependent variables, **Y**) into a few latent variables (LVs), which represent their linear combinations. The LVs are extracted in the direction of not only maximum variance in **X** and **Y** but also the maximum covariance between **X** and **Y** (i.e., correlation). The original matrices **X** and **Y** are linearly decomposed into:X = TPT + E(1)
Y = UQT + F(2)
where T and U represent matrices of scores (with the extracted LVs), P and Q represent matrices of loadings, whereas E and F represent the residual matrices. The LVs were originally computed using the conventional NIPALS algorithm [50]. However, in this work, the SIMPLS algorithm was used due to its simplicity and effectiveness [52]. In essence, both of these algorithms correlate X and Y through two sets of weights—W and C, and the following relationships:T = XW(3)
U = YC(4)
subject to w^T^w = 1, t^T^t = 1 for a maximum t^T^u.

Generally, the number of LVs is a parameter that requires careful optimization as not to over-fit the relationships between X and Y. As such, the number of LVs was optimized using LOO-CV. From the graphical depiction of root mean square error of CV (RMSECV) with respect to the number of LVs, the optimal number was determined from the knee point. RMSECV was defined as:(5)$$\mathrm{RMSECV}=\sqrt{\frac{\sum_{i=1}^{n}{\left(y\left(\mathrm{LOO}-\mathrm{CV}\right)-y\left(\exp.\right)\right)}^{2}}{n}}$$
where *y*(LOO-CV) represents the predicted retention times obtained through LOO-CV, *y*(exp.) represents experimentally-obtained retention time values, whereas *n* is the number of training analytes.

#### 3.5.2. Genetic Algorithms (GAs)

GAs are a family of global optimization algorithms that mimic the theory of evolution. First introduced by Holland [53,54], GAs have gained traction for solving various optimization problems; in this case, the non-polynomial hard (NP-hard) problem of variable selection. In a typical formulation of GA, an initial population of units is randomly generated and evolved through the generations in the direction of optimal fitness. A specified number of elite units survive to each next generation, while the other units are a result of cross-over (mating) of the remaining parent units. As in natural evolution, in GA, there is a possibility of mutation in each generation. For variable selection, the binary implementation of GA was used [47,48]. Binary encoding was employed to represent all the 13 molecular descriptors, with values of one or zero being assigned to selected or not selected variables. The objective function used for variable selection was RMSECV, defined with Equation (1) and calculated from the PLS model built out of the selected molecular descriptor subset. Parameters of GA-PLS were carefully optimized, as described in the following section.

#### 3.5.3. Optimization of GA-PLS Parameters

According to a previously published protocol [16], LOO-CV was used to optimize all the parameters of the GA-PLS algorithm employing a grid search. Namely, for GA, the population size was optimized in [10:10:20], the cross-over fraction in [0.1:0.1:0.8], the mutation rate in the same interval ([0.1:0.1:0.8]), whereas, for creation of the initial population, a uniform function was used. Furthermore, a uniform mutation, a single-point cross-over, and the tournament selection function were employed. The number of elite units was limited to 2, the number of maximum generations to 100, whereas the stall limit (the stopping criterion), i.e., the number of iterations without a change in RMSECV, was set to 10. All the GA parameters were optimized for the K-C18 column and applied for GA-PLS QSRR models of other chromatographic columns to preserve consistency. As for PLS, the number of latent variables was optimized using LOO-CV in (2:1:rank(**X**)) each GA unit.

#### 3.5.4. Consensus Modeling

For each chromatographic column, each GA-PLS run was repeated 1000 times, yielding 3000 QSRR models (1000 per column). Occurrence (% of selection) of the variables and its respective sample mean were calculated. Final GA-PLS QSRR models, termed consensus models, were built out of the variables with an occurrence higher than their respective sample mean.

#### 3.5.5. Molecular Descriptors for Mechanistic QSRR Modeling

Mechanistic QSRR modeling was carried out, employing molecular descriptors, describing: (i) hydrophobicity and the solvent effect, (ii) antioxidant activity, and (iii) electrostatic effects. Molecular descriptors used for GA-PLS QSRR models with brief descriptions are defined in Table 6 (values of QM parameters are given in Appendix A).

#### 3.5.6. Hydrophobicity and the Solvent Effect

Hydrophobicity, as the predominant effect of the reversed phase-HPLC (RP-HPLC) mechanism, was accounted for through the QM parameter solvation energy (SE). The mixture of the less polar analytes (such as flavonoids) in the polar mobile phase (such as acetonitrile-water) was accompanied by an increase in their SE. SE was defined as the difference between enthalpy-corrected energy values of the analytes in implicit water solvent (E(analyte)_in water solvent_) and in vacuo (E(analyte)*_in vacuo_*):SE = E(analyte)*_in water solvent_* − E(analyte)*_in vacuo_*(6)

#### 3.5.7. Antioxidant Activity

As previously mentioned, analytes separated on the three evaluated columns belong to the family of flavonoids, which are known for their antioxidant activity [55]. Besides hydrophobicity, parameters of two established antioxidant activity mechanisms (hydrogen atom transfer—HAT [43] and sequential proton-loss electron transfer—SPLET [56,57,58]) had, thereby, also been used for QSRR modeling. Both the considered mechanisms are schematically depicted in Figure 6. Parameters of the HAT mechanism were the number of hydroxyl groups (*n*(OH)) and minimum bond dissociation enthalpy (BDE_min_).

As for the SPLET mechanism, only the parameters of the first oxidation step: electron transfer enthalpy (ETE) and proton affinity (PA) were considered. Parameters of the second oxidation step were omitted from QSRR modeling because the diradical and diradical anion species are not energetically favorable and are unstable in solution [34]. BDE_min_, ETE, and PA were defined with the following equations:BDE = H(Ar(OH)n(O˙)) + H(H˙) − H(Ar(OH)(n + 1)(O˙))(7)
ETE = H(Ar(OH)n(O˙)) + H(e^−^) − H([Ar(OH)n(O˙)O:]^−^)(8)
PA = H[Ar(OH)n(O˙)] + H(H^+^) − H([Ar(OH)]n + 1)(9)
where E(H˙), H(e^-^), and H(H^+^) represent enthalpies of the hydrogen radical species H˙, electron e^-^, and proton H^+^, respectively. The values of these parameters were obtained from Refs. [34,59]. For in vacuo calculations, H(H^+^) and H(e^-^) were 1.481 kcal mol^−1^ and 0.752 kcal mol^−1^, respectively. In implicit (SMD) water solvent, H(H^+^) of −250.574 kcal mol^−1^ and H(e^-^) of −17.816 kcal mol^−1^ were used.

#### 3.5.8. Electrostatic Effects

One of the earliest and quite robust QM-based QSRR models introduced by Kaliszan’s group [11,44,45] included two parameters to account for electrostatic interactions between the analytes and the mobile and stationary phases. The two parameters were the total dipole moment (*Μ_tot._*) and the excess charge of the most negatively charged atom (*δ*_min_). The dipole moment of the analytes accounted for the dipole-dipole and dipole-induced dipole interactions with both the mobile and stationary phases. Stronger polar interactions, including ion-dipole interaction between the analytes and both chromatographic phases, were reflected through the *δ*_min_ parameter, which was calculated using the natural bond orbital (NBO) analysis [60,61].

Furthermore, in this work, the electrostatic contributions to the retention mechanism of flavonoids were expanded with the following important QM parameters: HOMO-LUMO energy gap (Δ*E*_HOMO-LUMO_), ionization potential (*IP*), electron affinity (*EA*), the global hardness (*η*), electronic chemical potential (*µ*), and electrophilicity (*ω*) [62,63].

Energies of the frontier orbitals (*E*(HOMO), *E*(LUMO)) of the analytes and their energy difference Δ*E*_HOMO-LUMO_ reflect their ability to donate or accept electrons. HOMO and LUMO energetics can give an indication of such reactivity of the analytes through approximations of *IP* and *EA* using the Koopmans’ theorem [64]. *IP* is approximated as negative HOMO energy, whereas *EA* is approximated as negative LUMO energy. Employing the same approximation, *η*, associated with the stability of the analytes, can be expressed as:(10)$$\eta =\frac{IP-EA}{2}=\frac{E\left(\mathrm{LUMO}\right)-E\left(\mathrm{HOMO}\right)}{2}$$ while *µ* and *ω* can be expressed as:
(11)$$µ=\frac{IP+EA}{2}=\frac{E\left(LUMO\right)-E\left(HOMO\right)}{2}$$
(12)$$\omega =\frac{1}{2}\frac{{µ}^{2}}{\eta }$$

Moreover, the negative of *µ* is considered to be absolute electronegativity. Smaller values of *µ* (larger values of electronegativity) are characteristic of electron acceptors, whereas electron donors exhibit large *µ* values (smaller values of electronegativity) [63]. Therefore, these parameters may account for the hydrogen bond interactions between the analytes and the chromatographic phases.

When defining parameters describing electrostatics using the Koopman’s theorem and the DFT HOMO-LUMO gap, one must take care of the so-called “band gap problem” [65,66], whereby the HOMO-LUMO gap is strongly under-estimated at the DFT level. Instead, physically it is more of an approximation of the excitation energy rather than I-A. However, in QSRR modeling, interest lies with the relative rather than absolute values of molecular descriptors, and correlations occur due to a cancellation of errors.

#### 3.5.9. Mechanistic QSRR Model Validation

For comprehensive validation of the mechanistic QSRR models, both leave-one-out cross-validation and external validation were employed.

#### 3.5.10. Leave-One-Out Cross-Validation (LOO-CV)

LOO-CV is an iterative internal validation technique in which one observation (in this case analyte) is left out, while the regression model is trained on the remaining ones. The process is repeated for *n* times until all the observations are exhausted.

In each iteration, a performance metric, such as RMSECV (defined with Equation (5)), is computed and averaged overall for a final LOO-CV performance.

#### 3.5.11. External Validation

As previously mentioned, the Kennard and Stone algorithm [46] was used to uniformly separate the dataset into training and (external) validation set. In such a manner, all the GA-PLS QSRR models were externally validated with new analytes. Besides LOO-CV performance, the final performance of both the training and validation sets was reported as sample mean RMSE (averaged over the training and testing sets):(13)$$\mathrm{RMSE}=\sqrt{\frac{\sum_{i=1}^{n}(y\left(\mathrm{pred}.\right)-y{\left(\exp.\right)}^{2}}{n}}$$
where *y*(pred.) represents the predicted retention times.

#### 3.5.12. Chemical Domain of Applicability

The chemical domain of applicability (AD) is a region in the **X**- and **Y**-spaces within the range of the predictive ability of the developed QSRR model. Typically, AD is depicted using a Williams plot, which represents a graphical depiction of the dependence between leverages (*h*) of analytes and standardized residuals. Leverages measure the distance of each analyte from the centroid of **X**-space and, as such, the influence of analytes on the QSRR model [67,68]. According to Atkinson [69], leverages are defined as the diagonal of the leverage matrix (**H**):(14)$$h=\mathrm{diag}(H)=\mathrm{diag}(X_{2}^{T}(X_{1}^{T}X_{1}){}^{-1}X_{2})$$
where **X**_1_ is the training set matrix of descriptors, whereas **X**_2_ can be either the training or validation set matrix of descriptors. AD is generally bounded by warning limits: (i) critical leverage value (*h**) and (ii) standard deviation of standardized residuals. Standardized residuals are typically bounded with two or three multiples of their respective standard deviation, whereas *h** is defined as [16]:(15)$$h^{*}=\frac{3\left(n+1\right)}{m}$$
where *n* represents the number of molecular descriptors, and *m* represents the number of observations of the training set.

## 4. Conclusions

In conclusion, a new approach based on GA-PLS QSRR modeling with the use of meaningful quantum-chemical descriptors was developed to shed light on the retention mechanisms for three chromatographic columns at the molecular level. The obtained GA-PLS models were physically interpretable due to the careful selection of molecular descriptors. Some distinct interactions affecting the retention of flavonoids were observed. For the IAM.PC.DD2 column, electrostatic interactions were found to be competing with solvent effects and hydrophobic forces. The π-π interactions for pentafluorophenyl column were suggested not to have an impact considering that the mobile phase was not fully comprised of acetonitrile.

The developed approach had the potential to serve as a starting point for a thorough analysis of interactions governing retention in HPLC and, thus, enhancing the selectivity for routine applications. Exemplified by our work, further research efforts should be directed towards the application of more realistic molecular descriptors to shed light on the retention mechanisms accounting for both strong and weak non-covalent interactions between the analytes and the chromatographic phases. Description of such mechanisms may provide valuable new insights for the assessment of biological activities of solutes using retention data obtained from stationary phases mimicking potential molecular targets, considering the importance of non-covalent interactions in biomolecular recognition.

## Figures and Tables

**Figure 1 ijms-21-02053-f001:**
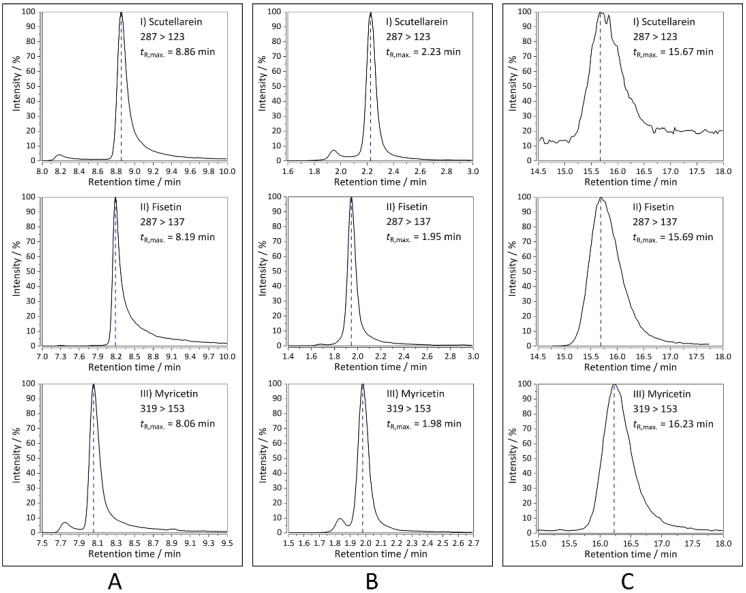
Representative multiple reaction monitoring (MRM) transitions and retention times as represented by scutellarein, fisetin, and myricetin, analyzed using HPLC-MS/MS on the: (**A**) K-C18, (**B**) K-F5, and (**C**) IAM.PC.DD2 chromatographic column.

**Figure 2 ijms-21-02053-f002:**
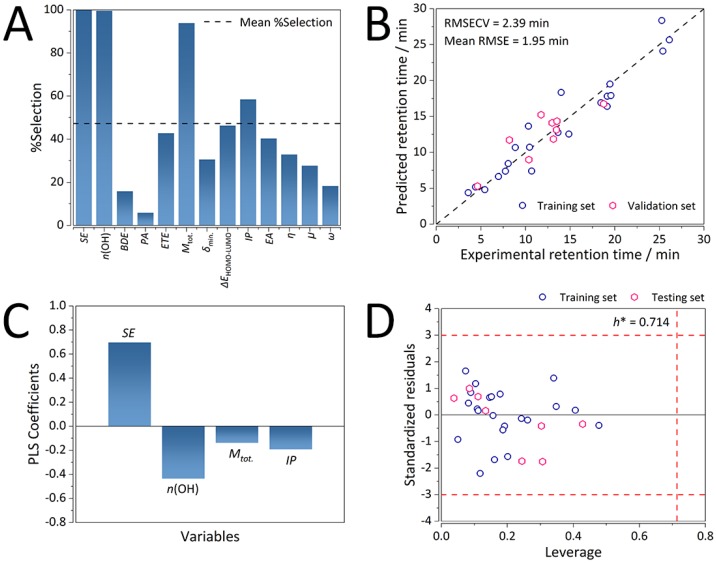
Performance characteristics of the consensus genetic algorithm-partial least squares (GA-PLS) quantitative structure retention relationships (QSRR) model for the C18 column. (**A**) Occurrence (expressed through % of selection) of molecular descriptor selection in 1000 GA-PLS runs. (**B**) Predictive ability on the training and validation sets (*n* = 30). (**C**) Distribution of the PLS coefficients (intercept = 0, due to autoscaling). (**D**) Applicability domain computed on the training and testing sets. Warning limits: three multiples of the standard deviation of standardized residuals, and critical leverage (*h**) of 0.714 (*n* = 30). Royal blue circles depict the training set observations, whereas the pink diamonds depict the testing set observations.

**Figure 3 ijms-21-02053-f003:**
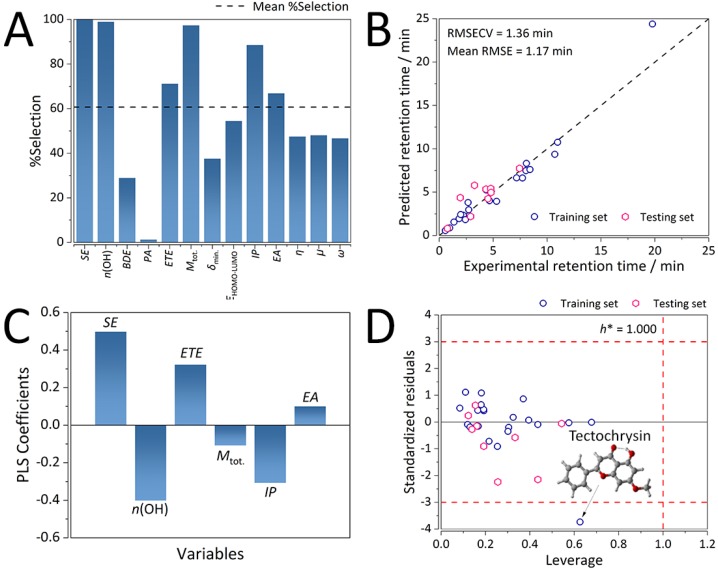
Performance characteristics of the consensus GA-PLS QSRR model for the K-F5 column. (**A**) Occurrence (expressed through % of selection) of molecular descriptor selection in 1000 GA-PLS runs. (**B**) Predictive ability on the training and validation sets (*n* = 30). (**C**) Distribution of the PLS coefficients (intercept = 0, due to autoscaling). (**D**) Applicability domain computed on the training and testing sets. Warning limits: three multiples of the standard deviation of standardized residuals, and critical leverage (*h**) of 1.000 (*n* = 30). Royal blue circles depict the training set observations, whereas the pink diamonds depict the testing set observations.

**Figure 4 ijms-21-02053-f004:**
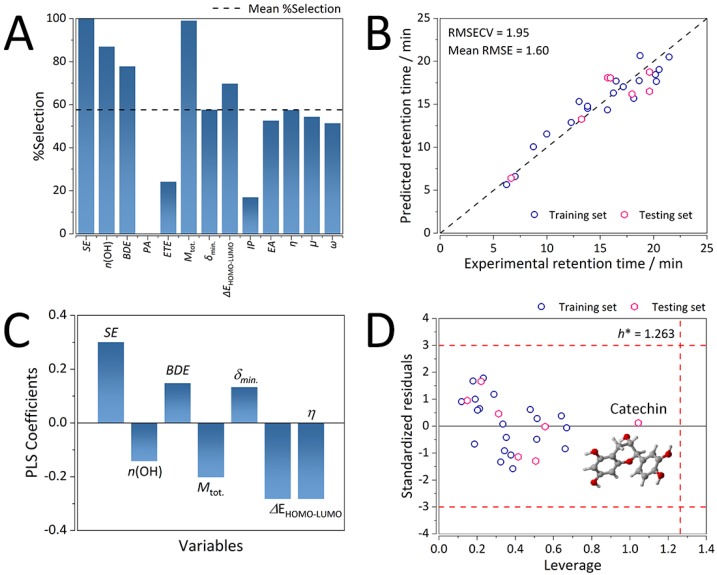
Performance characteristics of the consensus GA-PLS QSRR model for the IAM.PC.DD2 column. (**A**) Occurrence (expressed through % of selection) of molecular descriptor selection in 1000 GA-PLS runs. (**B**) Predictive ability on the training and validation sets (*n* = 27). (**C**) Distribution of the PLS coefficients (intercept = 0, due to autoscaling). (**D**) Applicability domain computed on the training and testing sets (n = 27). Warning limits: three multiples of the standard deviation of standardized residuals, and critical leverage (*h**) of 1.263. Royal blue circles depict the training set observations, whereas the pink diamonds depict the testing set observations.

**Figure 5 ijms-21-02053-f005:**
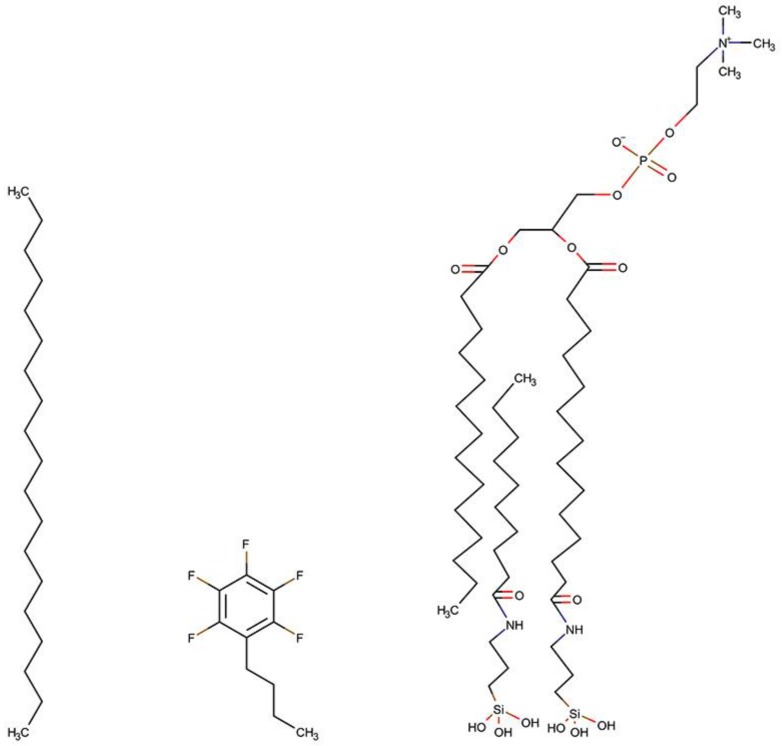
Structures of the ligands chemically-bonded to the three stationary phases.

**Figure 6 ijms-21-02053-f006:**
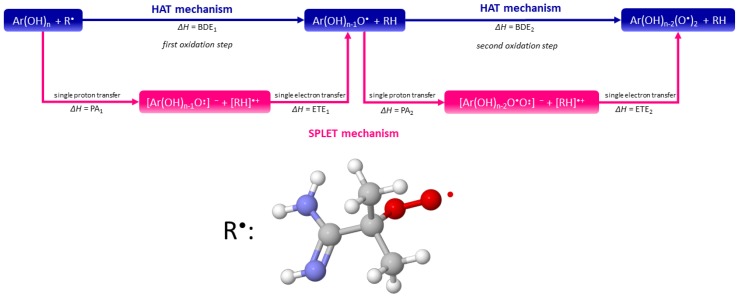
Schematic representation of the hydrogen atom transfer (HAT) and sequential proton-loss electron transfer (SPLET) mechanisms (adapted from Appendix A in Ref. [34]).

**Table 1 ijms-21-02053-t001:** Univariate retention time correlation matrix for the three evaluated columns.

Retention	Coefficients	t_R_ (K-C18)	t_R_ (K-F5)	t_R_ (IAM.PC.DD2)
t_R_ (K-C18)	R	1		
	p	n.a.		
t_R_ (K-F5)	R	0.93	1	
	p	1.42 × 10^−13^	n.a.	
t_R_ (IAM.PC.DD2)	R	0.81	0.79	1
	p	2.81 × 10^−7^	1.22 × 10^−6^	n.a.

t_R_—retention times for each columns: K-C18, K-F5 and IAM.PC.DD2.

**Table 2 ijms-21-02053-t002:** Statistical significance of the consensus genetic algorithm-partial least squares (GA-PLS) model for the K-C18 column.

Source	SS	df	MS	F	Prob. > F
Total	19.33	20	0.966	36.27	1.27 × 10^−4^
Fit	18.15	6	3.024		
Residual	1.18	14	0.083		

*SS*—sum of squares, *df*—degrees of freedom, *MS*—mean square.

**Table 3 ijms-21-02053-t003:** Statistical significance of the consensus GA-PLS model for the K-F5 column.

Source	SS	df	MS	F	Prob. > F
Total	19.36	20	0.97	49.84	3.6 × 10^−6^
Fit	18.81	8	2.35		
Residual	0.55	12	0.05		

*SS*—sum of squares, *df*—degrees of freedom, *MS*—mean square.

**Table 4 ijms-21-02053-t004:** Statistical significance of the consensus GA-PLS model for the IAM.PC.DD2. column.

Source	SS	df	MS	F	Prob. > F
Total	17.46	18	0.97	39.39	1.05 × 10^−4^
Fit	16.75	6	2.79		
Residual	0.71	12	0.07		

*SS*—sum of squares, *df*—degrees of freedom, *MS*—mean square.

**Table 5 ijms-21-02053-t005:** Physicochemical parameters of the evaluated chromatographic columns.

#	Column Name	Length / mm	Internal Diameter (ID) / mm	Particle Size /μm	Carbon Load / %	Pore Size / Å	Surface Area / m^−2^ g	Ligand Type *	Surface coverage density (α_RP_) / µmol/m^2^ **
1	K-C18	150	4.6	5	12	100	200	C18	3.23
2	K-F5	100	2.1	2.6	9	100	200	C-F5	5.11
3	IAM.PC.DD2	150	4.6	10	7	300	110	diacylated PC	1.53

* C18—octadecyl, C-F5—pentafluorophenyl, PC—phosphatidylcholine. ** *α_RP_* was calculated according to the Berendsen-de-Galan equation [23].

**Table 6 ijms-21-02053-t006:** Molecular descriptors used for GA-PLS quantitative structure retention relationships (QSRR) models.

Name	Description
Solvation energy (SE)	defined in Equation (6)
Number of hydroxyl groups (n(OH))	number of OH-groups in flavonoid structure
Minimum bond dissociation enthalpy (BDE_min_)	parameter of the first oxidation step of SPLET mechanism, defined in Equation (7)
Proton affinity (PA)	PA is the negative quantity of proton-gain enthalpy, which is a standard enthalpy of the reaction: A^−^ _(g)_ +H^+^_(g)_ → HA_(g)_
Electron transfer enthalpy (ETE)	parameter of the first oxidation step of SPLET mechanism, defined in Equation (8)
Excess charge of the most negatively charged atom (*δ*_min_)	shows the ability of analytes to participate in polar interactions with the phases of the charge transfer and hydrogen bonding
Total dipole moment M_tot._	accounts for the dipole-dipole and dipole-induced dipole attractive interactions of the analyte with mobile and stationary phases
**HOMO-LUMO energy gap** (Δ**E_HOMO-LUMO_**)	the difference between the HOMO and LUMO energies GAP = ɛ _LUMO_ − ɛ _HOMO_, where ɛ _LUMO_ and ɛ _HOMO_ are the energies of the lowest unoccupied molecular orbital and the highest occupied molecular orbital, respectively
Ionization potential (IP)	ionization potential (or ionization energy) is defined as the energy needed to extract one electron from a chemical system, i.e., IP = E(N_el_) − E (N_el_ − 1), where N_el_ is the number of electrons of the system
Electronic chemical potential (µ)	negative of electronegativity
Electrophilicity (ω)	electrophilicity can be defined as ω = (E_HOMO_+E_LUMO_)^2^/2(E_LUMO_ − E_HOMO_)
Global hardness (η)	can be defined as resistance to charge transfer, Equation (10)
Electron affinity (EA)	EA is the energy released when an electron attaches to a gas-phase atom: E_(g)_ + e^−^_(g)_ → E^−^_(g)_

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
