# Peer review of "Mechanistic Chromatographic Column Characterization for the Analysis of Flavonoids Using Quantitative Structure-Retention Relationships Based on Density Functional Theory"

_ijms, 2020, doi:10.3390/ijms21062053_

Round 1
Reviewer 1 Report
The manuscript by B. Buszewski et al. is focused on QSRR method for characterization of three different columns used for flavonoids retention. The manuscript brings novel approach to separation system characterization. Nevertheless, the manuscript needs moderate revision before publication.
Comments:
- The title of the manuscript is slightly misleading. The columns were rather used for “flavonoids retention” not separation as you did not optimize separation because of using MRM.
- Some discussion/correlation of flavonoids retention and structure with QSRR results should be added.
- The choice of these three columns should be specified. Moreover, the columns of different dimensions and particle sizes were used.
- Page 5, line 150: correct the variable, there should be SE instead of BDEmin
- Page 6, line 189: Figure 2D instead of 4D.
- Materials and Methods: Column temperature is missing.
- Page 17, lines 594-596: please reformulate, this makes no sense compared to the text (page 8, lines 243-249).
Author Response
We are very grateful to your critical comments and thoughtful suggestions. Based on these comments and suggestions, we have made a careful revision of the original manuscript. We would like to thank both the editor and the reviewers who made contributions that led to improvement of our paper.
Responses to reviewer 1 comments:
Point 1: The title of the manuscript is slightly misleading. The columns were rather used for “flavonoids retention” not separation as you did not optimize separation because of using MRM.
Response 1: We agree with the reviewer that the title is somewhat misleading. In this work, columns were used for retention of flavonoids for construction of GA-PLS QSRR models. The title was revised to 'Mechanistic chromatographic column characterization for the analysis of flavonoids using quantitative structure-retention relationships based on density functional theory'.
Point 2: Some discussion/correlation of flavonoids retention and structure with QSRR results should be added.
Response 2: We thank the reviewer for the suggestion. Being a mature technique, strong correlations and interpretations of the underlying relationships between retention and structure is expected for QSRR models. However, descriptors used for construction of the models already reflect structural features of analytes and Figures 2-4 demonstrate their contributions to the QSRR models with relevant interpretation. Furthermore, there are such discussions in the sections dedicated to mechanistic interpretations of GA-PLS models for each of the columns.
Point 3: The choice of these three columns should be specified. Moreover, the columns of different dimensions and particle sizes were used.
Response 3: Three columns were selected since they are often used for separation of flavonoids and that gives this work practical dimension. The structure of IAM.PC.DD2 column was interesting to consider owing to its dual nature comprising both of hydrophobic parts and charged centers. The IAM.PC.DD2 column is widely used in biomimetic chromatography and valuable insights into its interactions with flavonoids were gained. In addition, the reviewer asked about the application of three different columns with different dimensions and particle sizes and the question is certainly reasonable. Three models developed for each column were treated as independent and individual parameters for all three columns were considered for each model.
Point 4: Page 5, line 150: correct the variable, there should be SE instead of BDEmin
Response 4: BDEmin was corrected to SE on line 148.
Point 5: Page 6, line 189: Figure 2D instead of 4D.
Response 5: Thank you for this comment. The error has been corrected.
Point 6: Materials and Methods: Column temperature is missing.
Response 6: The column temperature was 30°C. This information was added to the text on line 382.
Point 7: Page 17, lines 594-596: please reformulate, this makes no sense compared to the text (page 8, lines 243-249).
Response 7: The sentence in lines 594-596 was re-formulated as recommended by the reviewer.
Reviewer 2 Report
This work is devoted to building and validating a QSRR model that predicts retention times of 30 structurally different flavonoids on three chromatographic columns. The authors employed the genetic algorithm-partial least square (GA-PLS) approach to obtain the consensus GA-PL QSRR model. Their QSRR model employs 13 quantum chemical descriptors of well-defined physical meaning which enables the retention mechanism of the studied compounds to be unrevealed. Thus, they demonstrated that for the K-C18 column hydrophobicity and solvent effects are dominating while electrostatic governs retention times for IAM.PC.DD2.
This is an interesting and well written paper that should be published. I have only several minor remarks which the authors might want to address preparing the final version of the article.
- Conformational search was based on the following optimization sequence: AM1 (all configurations) à HF/3-21G (20 conformations), ωb97xD/6-311++G(d,p) (5 conformations). What was the reason for the HF optimizations? The energy accuracy of HF method is lower than that of AM1 (AM1 includes correlation energy via parameterization while HF does not account for correlation effects at all). Therefore, it seems that the optimization sequence proposed by the authors may leave out low energy structure(s).
- The authors assume that the DFT HOMO-LUMO gap is the approximation to the I – A difference (ionization energy – electron affinity). If HF HOMO-LUMO is indeed such an approximation to I-A, the situation is quite different for DFT HOMO-LUMO. Indeed, it is frequently called ‘‘the band gap problem’’ of DFT (Phys.Chem. Chem. Phys., 2013, 15, 16408; J. Chem. Theory Comput. 2017, 13, 1656−1666) – the band gap is strongly underestimated at the DFT level and physically it is approximation to the excitation energy rather than to I-A. The authors should at least mention this problem.
- The QM descriptors specific for the antioxidant properties (SPLET and HAT mechanism) of flavonoids are included in the QSRR model. However, a relation between antioxidant properties and retention times are not straightforward (at least to me). The authors might want to comment on the necessity of using BDE, ETE, and PA in their chemometric model.
- Lack of the table explaining all 13 descriptors makes reading the text, especially figures 2-4, difficult. Therefore, I strongly recommend addition of such a table.
Author Response
We are very grateful to your critical comments and thoughtful suggestions. Based on these comments and suggestions, we have made a careful revision of the original manuscript. We would like to thank both the editor and the reviewers who made contributions that led to improvement of our paper.
Responses to reviewer 2 comments:
This work is devoted to building and validating a QSRR model that predicts retention times of 30 structurally different flavonoids on three chromatographic columns. The authors employed the genetic algorithm-partial least square (GA-PLS) approach to obtain the consensus GA-PL QSRR model. Their QSRR model employs 13 quantum chemical descriptors of well-defined physical meaning which enables the retention mechanism of the studied compounds to be unrevealed. Thus, they demonstrated that for the K-C18 column hydrophobicity and solvent effects are dominating while electrostatic governs retention times for IAM.PC.DD2.
This is an interesting and well written paper that should be published. I have only several minor remarks which the authors might want to address preparing the final version of the article.
Point 1: Conformational search was based on the following optimization sequence: AM1 (all configurations) à HF/3-21G (20 conformations), ωb97xD/6-311++G(d,p) (5 conformations). What was the reason for the HF optimizations? The energy accuracy of HF method is lower than that of AM1 (AM1 includes correlation energy via parameterization while HF does not account for correlation effects at all). Therefore, it seems that the optimization sequence proposed by the authors may leave out low energy structure(s).
Response 1: The reviewer raises an important concern and is correct to a certain extent. In the present study, AM1 was used to cover a large conformational space, and HF/3-21G was used to filter out the top 20 conformations. We assumed that the AM1 method may not have been parametrized for more complex flavonoids which exhibit stronger non-bonded pi-pi interactions. For simpler more planar analytes, both HF/3-21G and AM1 are expected to yield similar results since the flavonoid molecules are dominated by hydrogen bonds which are well described by the HF theory. We have confirmed our assumption based on benchmark calculations of quercetin.
Point 2: The authors assume that the DFT HOMO-LUMO gap is the approximation to the I – A difference (ionization energy – electron affinity). If HF HOMO-LUMO is indeed such an approximation to I-A, the situation is quite different for DFT HOMO-LUMO. Indeed, it is frequently called ‘‘the band gap problem’’ of DFT (Phys. Chem. Chem. Phys., 2013, 15, 16408; J. Chem. Theory Comput. 2017, 13, 1656−1666) – the band gap is strongly underestimated at the DFT level and physically it is approximation to the excitation energy rather than to I-A. The authors should at least mention this problem.
Response 2: The reviewer raises a valid point. When defining parameters describing electrostatics using the Koopman’s theorem and the calculated DFT HOMO-LUMO, one must take care of the so-called “band gap problem” whereby the HOMO-LUMO gap is strongly underestimated at the DFT level. Instead, physically it is more of an approximation of the excitation energy rather than I-A. However, in QSRR modelling interest lies with the relative rather than absolute values of molecular descriptors, and correlations occur due to a cancellation of errors. To highlight the potential issue, the main text was updated with a brief discussion, whereas the two mentioned references were also included.
Point 3: The QM descriptors specific for the antioxidant properties (SPLET and HAT mechanism) of flavonoids are included in the QSRR model. However, a relation between antioxidant properties and retention times are not straightforward (at least to me). The authors might want to comment on the necessity of using BDE, ETE, and PA in their chemometric model.
Response 3: Parameters specific for the antioxidant properties of flavonoids were used since they relate to their molecular structures and thus their effect on interactions with stationary phases and in turn on retention.
Point 4: Lack of the table explaining all 13 descriptors makes reading the text, especially figures 2-4, difficult. Therefore, I strongly recommend addition of such a table.
Response 4: For clarity, a table with all descriptors and brief descriptions is included in the main text as Table 6.
Reviewer 3 Report
The paper reports novel QSRR models for flavonoids using quantum-based molecular descriptors. Models are developed using genetics algorithms. To the best of my knowledge, the results are new and correct. Also, they are presented in a rather clear and understandable manner. Therefore, I recommend the acceptance of this article in its present form.
Author Response
We are very grateful to your critical comments and thoughtful suggestions. Based on these comments and suggestions, we have made a careful revision of the original manuscript. We would like to thank both the editor and the reviewers who made contributions that led to improvement of our paper.
The paper reports novel QSRR models for flavonoids using quantum-based molecular descriptors. Models are developed using genetics algorithms. To the best of my knowledge, the results are new and correct. Also, they are presented in a rather clear and understandable manner. Therefore, I recommend the acceptance of this article in its present form.
Response: We thank the reviewer for recognizing the novelty of our work and appreciate his/her participation in the review process of this manuscript.